# High rate of critical coronary stenosis in comatose patients with Non-ST-elevation out-of-hospital cardiac arrest (NSTE-OHCA) undergoing therapeutic hypothermia— Experience from the HAnnover COoling REgistry (HACORE)

**Vera Garcheva**[☯], **Muharrem Akin**[iD][☯], **John Adel, Carolina Sanchez Martinez, Johann Bauersachs, Andreas Schäfer**[iD]*

Department of Cardiology and Angiology, Medizinische Hochschule Hannover, Hannover, Germany

☯ These authors contributed equally to this work.
* Schaefer.andreas@mh-hannover.de

## Abstract

### Background

Myocardial infarction is the most frequent cause for out-of-hospital cardiac arrest (OHCA) in adults. Patients with ST-segment elevations (STE) following return of spontaneous circulation (ROSC) are regularly admitted to the catheterisation laboratory for urgent coronary angiography. Whether patients without obvious STE (NSTE) should receive coronary angiography as part of a standardised diagnostic work-up following OHCA is still debated.

### Methods

We analysed a cohort of 517 subsequent OHCA patients admitted at our institution who received a standardised diagnostic work-up including coronary angiography and therapeutic hypothermia. Patients were 63±14 years old, 76% were male. Overall, 180 (35%) had ST-elevation in the post-ROSC ECG, 317 (61%) had shockable rhythm (ventricular fibrillation or tachycardia) at first ECG. ROSC was achieved after 26±21 minutes.

### Results

Critical coronary stenosis requiring PCI was present in 83% of shockable and 87% of non-shockable STE-OHCA and in 48% of shockable and 22% of non-shockable NSTE-OHCA patients. In-hospital survival was 61% in shockable and 55% in non-shockable STE-OHCA and 60% in shockable and 28% in non-shockable NSTE-OHCA.

### Conclusion

Standardised admission diagnostics in OHCA patients undergoing therapeutic hypothermia with a strict admission protocol incorporating ECG and coronary catheterisation shows a

**Data Availability Statement:** General public deposition is restricted by data protection law. Regarding data availability there is a legal restriction as the patient sample includes potentially identifying patient information and specific dates. The datasets used and/or analysed during the current study are available from the ethics committee on reasonable request. Contact information for the ethics committee is: Ethikkommission@mh-hannover.de.

**Funding:** The study was partly supported by the Clinical Research Group (KFO) 311 of the Deutsche Forschungsgemeinschaft to JB. There was no additional external or internal funding received for this study.

**Competing interests:** AS received modest lecture fees from ZOLL Inc. regarding therapeutic hypothermia. All other authors have no conflict of interest to declare. This does not alter our adherence to PLOS ONE policies on sharing data and materials.

high rate of relevant coronary stenosis in STE-OHCA irrespective of the initial rhythm and in NSTE-OHCA with initial shockable rhythm. Based on the unfavourable outcome and low PCI rate observed in NSTE-OHCA patients with a primary non-shockable ECG rhythm it might be reasonable to restrict routine early coronary angiography to patients with primary shockable rhythms and/or ST-segment elevations after ROSC.

## Introduction

After out-of-hospital cardiac arrest (OHCA) cardio-pulmonary resuscitation (CPR) is provided to get return of spontaneous circulation (ROSC) as quickly as possible [1] and to prevent cerebral reperfusion injury [2]. The majority of cardiac arrest cases is attributed to cardiac causes with acute myocardial infarction and arrhythmias in patients with underlying heart disease being the most frequent ones [1]. In the landmark TTM-trial, a large proportion of patients with presumed cardiac cause of arrest died early due to evolving haemodynamic instability and coronary angiography had not been performed routinely [3]. Urgent coronary angiography is well recommended for patients with ST-segment elevations in their post-ROSC electrocardiogram (ECG), but there is no clear evidence for OHCA patients without ST-segment elevation (non-STE = NSTE) [1]. While urgent coronary angiography is highly recommended for NSTE-myocardial infarction patients with subsequent cardiac arrest [4,5], it is difficult to diagnose NSTE-myocardial infarction in patients with a primary presentation as OHCA, because elevated biomarkers such as troponin following CPR are not specific to proof myocardial infarction as the cause of cardiac arrest.

It remains unclear whether all NSTE-OHCA patients with a presumed cardiac cause for arrest should undergo routine coronary angiography as part of a standardised diagnostic work-up. We intend to provide an uninterrupted phase of intensive care including therapeutic hypothermia following ICU admission by performing potentially relevant diagnostic procedures early before ICU admission. The recent randomised COACT trial testing the concept of immediate angiography in patients with an initial shockable rhythm but absence of ST-segment elevations showed no survival benefit by early invasive assessment [6].

We previously adopted a strategy in our hospital of interdisciplinary screening in the emergency room including endotracheal airway management and early assessment of ventricular function by transthoracic echo [7]. All patients without evident non-cardiac cause for arrest undergo cardiac catheterisation and percutaneous coronary intervention (PCI) if needed, intravascular cooling, and placement of active hemodynamic assist devices if patients are in shock [8,9]. At latest after cardiac catheterization, all OHCA patients receive a cerebral and thoracic computed tomography as a routine workup before admission to our cardiology intensive care unit (ICU) harbouring the cardiac arrest centre, where therapeutic hypothermia and continuous neuromonitoring are initiated immediately upon arrival in all OHCA patients [10].

Here we report the rate of PCI and resulting in-hospital survival based on the presence of a shockable vs non-shockable rhythm during first post-arrest ECG in combination with the presence or absence of ST-segment elevations in ECGs documented either by ambulance or in the emergency room with the intention of using those early ECG markers as potential guidance for further diagnostic approaches.

## Methods

### Study design

The HAnnover COoling REgistry (HACORE) is prospective observational and in accordance with the Declaration of Helsinki and approved by the ethics committee (#3567–2017) at

Hannover Medical School. The ethics committee approved the analysis as reported in the present manuscript. Written informed consent was obtained from legal guardians during the unconscious period and re-consented by survivors after gaining consciousness. HACORE includes anonymized data from all OHCA patients treated at our cardiac arrest centre with a standardized protocol including therapeutic hypothermia. Here, all patients receiving therapeutic hypothermia following OHCA were analysed with regard to their presenting ECG patterns and coronary angiography results.

## Patient population

Consecutive comatose OHCA patients (n = 517) with presumed cardiac cause of arrest who received therapeutic hypothermia between January 1st 2011 and June 30st 2019 served as cases. All patients were admitted to the ICU at the Department of Cardiology and Angiology at Hannover Medical School and treated according to an institutional protocol ensuring a standardized approach including early diagnostics by computed tomography and coronary angiography as well as early haemodynamic stabilization in case of cardiogenic shock using microaxial pumps and/or extracorporeal life support [7]. Microaxial pumps were used primarily in isolated left-ventricular failure whereas ECMO was used in refractory arrest or biventricular failure. All OHCA patients with presumed cardiac cause of arrest are mandatorily treated by this protocol in order to provide optimal guideline-recommended therapy and, therefore, receive therapeutic hypothermia [7–9,11–13].

## Patient treatment

Patients after primarily successful cardio-pulmonary resuscitation were first screened and stabilized in the emergency room. After initial assessment, all STE-OHCA patients were taken to the catheterisation laboratory. If non-cardiac causes such as asphyxia, stroke, intracranial bleeding, strangulation, drowning, pulmonary embolism or aortic dissection were suspected in NSTE-OHCA during initial assessment, patients received computed tomography first. If none of those suspicions were raised, NSTE-OHCA patients were treated like STE-OHCA and underwent coronary angiography first. In concomitant cardiogenic shock, active haemodynamic support with an Impella micro-axial pump was initiated as standard procedure in the catheterisation laboratory.

## Clinical follow-up

Patients were followed up for the time period of their hospital stay.

## Statistical analysis

Numbers are given as n (%), means ± standard deviation (SD) for quantification, or median and interquartile ranges (IQR) in the tables. Statistical analysis was performed with ANOVA and Mann-Whitney test as nonparametric test followed by a Bonferroni test for multiple comparisons. Chi-square test was applied to compare patient characteristics. Cumulative mortality was estimated by Kaplan-Meier method and compared by the log-rank test. Data were analysed using SPSS Statistics 24 (IBM SPSS Statistics 24). A two-sided P-value of < 0.05 was considered statistically significant.

While the primary analysis focused on the presence and absence of shockable rhythms and on the presence or absence of ST-segment elevation, a second step of the analysis was focused on the subgroup of patients in our registry, who matched the inclusion and exclusion criteria of the COACT trial, which investigated the effect of immediate compared to delayed coronary angiography in NSTE-OHCA [6].

## Results

### Patient characteristics

The overall OHCA patient population receiving therapeutic hypothermia consisted of 517 consecutive patients. ROSC had been achieved after 26±21 minutes. The majority of patients (61%) had a primarily shockable rhythm defined as either ventricular tachycardia or ventricular fibrillation. In 28 patients (5%), extracorporeal CPR had to be initiated upon arrival using vaECMO [11]. In 44 (9%) patients, coronary angiography was not performed due to identification of a non-cardiac cause of arrest during primary assessment. Critical coronary stenosis requiring PCI was present in 151 (84%) STE-OHCA and in 118 (40%) NSTE-OHCA patients (Table 1). Overall, 116 (23%) OHCA patients required mechanical support for cardiogenic shock, with 37 (32% of patients with circulatory support) being supported by vaECMO plus Impella [8].

### The role of ST-segment elevations after ROSC

Patients with STE-OHCA were younger, less comorbid, had a higher rate of shockable rhythm at first rhythm evaluation, and better renal function (Table 1). As expected, STE-OHCA patients were more likely to require PCI during coronary angiography, had less extensive coronary artery disease predominantly located in the LAD, and required mechanical circulatory support more often than NSTE-OHCA patients (Table 2). STE-OHCA patients had a higher in-hospital survival rate than NSTE-OHCA patients. Comparing PCI and non-PCI group within the NSTE-OHCA cohort, only witnessed arrest and primary shockable rhythm were identified by univariate analysis (S1 Table).

When applying the inclusion and exclusion criteria from the COACT trial to our population (only OHCA with initially shockable rhythm remaining unconscious after ROSC without STE, shock or an obvious non-coronary cause of arrest [6]) we found that only a minority of 102 out of 337 (30%) of our daily-practice NSTE-OHCA patients would have fit the trials' criteria. These very selected patients had an in-hospital mortality rate of 31% in our registry comparable to the reported 35% in the angiography group within that trial (OR 0.86 95%-CI 0.53–1.39) [6].

### The role of a shockable rhythm at first ECG

Patients with shockable rhythms were younger, had fewer history of cerebrovascular or obstructive pulmonary disease, higher rates of witnessed arrest, lower lactate levels on admission, better renal function, and lower NSE levels at day 3 compared to patients with non-shockable rhythms (Table 1). Patients with shockable rhythms were more likely to undergo coronary angiography and to require PCI, had more extensive coronary artery disease, and required mechanical circulatory support more often than patients with non-shockable rhythms. They had a higher in-hospital survival rate than patients with non-shockable rhythms (Table 2). Comparing PCI and non-PCI group within the non-shockable OHCA cohort, only a higher troponin level on admission was identified by univariate analysis (S1 Table).

### Influence of combined ECG patterns on PCI rate and in-hospital survival

Critical coronary stenosis requiring PCI was present in 83% of shockable and 87% of non-shockable STE-OHCA and in 48% of shockable and 22% of non-shockable NSTE-OHCA patients (Fig 1).

In-hospital survival was 61% in shockable and 55% in non-shockable STE-OHCA and 60% in shockable and 28% in non-shockable NSTE-OHCA (Fig 2), respectively. The difference on

**Table 1. Baseline characteristics.**

| | STE-OHCA | | NSTE-OHCA | | p value | Shockable rhythm | | Non-shockable rhythm | | p value |
|---|---|---|---|---|---|---|---|---|---|---|
| Number (%) | 180 | (35) | 337 | (65) | | 317 | (61) | 200 | (39) | |
| Age–years | 61±12 | | 65 ±15 | | 0.001 | 61±14 | | 67±14 | | <0.001 |
| Male sex, n (%) | 147 | (82) | 247 | (73) | 0.040 | 256 | (81) | 138 | (69) | 0.030 |
| In-hospital survival (%) | 119 | (66) | 167 | (50) | <0.001 | 208 | (66) | 78 | (39) | <0.001 |
| **Cardiovascular risk factors** | | | | | | | | | | |
| Hypertension (%) | 94 | (52) | 191 | (57) | 0.350 | 175 | (55) | 110 | (55) | 1.000 |
| Diabetes (%) | 30 | (17) | 86 | (26) | 0.049 | 58 | (18) | 58 | (29) | 0.005 |
| Hyperlipidaemia (%) | 72 | (40) | 104 | (31) | 0.041 | 110 | (35) | 66 | (33) | 0.704 |
| Family history for CAD (%) | 15 | (8) | 25 | (7) | 0.730 | 34 | (11) | 6 | (3) | 0.001 |
| Smoking (%) | 78 | (43) | 87 | (26) | <0.001 | 118 | (37) | 47 | (24) | 0.010 |
| **Previous comorbidities** | | | | | | | | | | |
| CAD (%) | 37 | (21) | 94 | (28) | 0.070 | 77 | (24) | 54 | (27) | 0.534 |
| PCI (%) | 20 | (11) | 39 | (12) | 1.000 | 37 | (12) | 22 | (11) | 0.887 |
| CABG (%) | 7 | (4) | 46 | (14) | <0.001 | 32 | (10) | 21 | (11) | 0.883 |
| PAD (%) | 11 | (6) | 31 | (9) | 0.240 | 20 | (6) | 22 | (11) | 0.069 |
| TIA/stroke (%) | 13 | (7) | 42 | (12) | 0.070 | 26 | (8) | 29 | (15) | 0.028 |
| CKD (%) | 13 | (7) | 52 | (16) | 0.080 | 34 | (11) | 31 | (16) | 0.134 |
| chronic RRT (%) | 1 | (1) | 5 | (1) | 0.600 | 2 | (1) | 4 | (2) | 0.212 |
| Atrial fibrillation (%) | 14 | (8) | 87 | (26) | <0.001 | 62 | (20) | 39 | (20) | 1.000 |
| Pacemaker (%) | 3 | (2) | 13 | (4) | 0.200 | 11 | (3) | 5 | (3) | 0.611 |
| ICD (%) | 2 | (1) | 4 | (1) | 0.300 | 2 | (1) | 4 | (2) | 0.119 |
| COPD/ Asthma (%) | 11 | (6) | 45 | (13) | 0.010 | 23 | (7) | 33 | (17) | 0.010 |
| **Characteristics of cardiac arrest** | | | | | | | | | | |
| Witnessed arrest (%) | 150 | (83) | 259 | (77) | 0.090 | 267 | (84) | 142 | (71) | <0.001 |
| Bystander CPR (%) | 128 | (71) | 214 | (64) | 0.100 | 227 | (72) | 115 | (58) | 0.010 |
| Shockable Rhythm (%) | 149 | (83) | 168 | (50) | <0.001 | - | - | - | - | - |
| ST-segment elevation (%) | - | - | - | - | - | 149 | (47) | 31 | (16) | <0.001 |
| ROSC, min | 28±21 | | 26±21 | | 0.250 | 27±21 | | 25±21 | | 0.230 |
| Ongoing CPR at admission (%) | 21 | (12) | 39 | (12) | 1.000 | 31 | (10) | 29 | (15) | 0.120 |
| eCPR (%) | 15 | (8) | 13 | (4) | 0.410 | 18 | (6) | 10 | (5) | 0.843 |
| Impella (%) | 49 | (27) | 46 | (16) | <0.001 | 72 | (23) | 23 | (14) | 0.020 |
| va ECMO (%) | 29 | (16) | 29 | (10) | 0.010 | 38 | (12) | 20 | (12) | 0.568 |
| Renal replacement therapy (%) | 44 | (24) | 104 | (35) | 0.127 | 84 | (27) | 64 | (40) | 0.194 |
| **Baseline laboratory values** | | | | | | | | | | |
| Lactate, mmol/l | 7.72±4.87 | | 8.22±4.39 | | 0.250 | 7.34±4.49 | | 9.18±4.47 | | <0.001 |
| pH | 7.16±0.19 | | 7.14±0.18 | | 0.270 | 7.18±0.16 | | 7.09±0.1.9 | | <0.001 |
| Creatinine, µmol/l | 108±48 | | 142±120 | | <0.001 | 111±54 | | 161±145 | | <0.001 |
| Urea nitrogen, mmol/l | 6.99±1.98 | | 7.81±3.73 | | 0.480 | 6.66±2.53 | | 8.93±3.98 | | 0.022 |
| Creatinkinase, U/l | 108±48 | | 142±120 | | 0.580 | 518±1328 | | 446±1026 | | 0.522 |
| hs-Troponine T, µg/l | 790±2619 | | 652±5004 | | 0.740 | 574±2202 | | 903±6397 | | 0.414 |
| NT-proBNP, ng/l | 806±2180 | | 937±2873 | | 0.700 | 706±1742 | | 1138±3507 | | 0.225 |
| Haemoglobin, g/dl | 13.57±1.99 | | 12.70±2.74 | | 0.004 | 13.55±1.88 | | 12.31±3.04 | | <0.001 |
| Leukocytes, *1000/µl | 14.93±6.55 | | 15.03±7.85 | | 0.920 | 14.79±7.63 | | 15.24±7.22 | | 0.630 |
| NSE- day 3, µg/l | 34 [21–76] | | 29 [19–52] | | 0.630 | 27 [20–46] | | 39 [20–117] | | 0.027 |

(*Continued*)

**Table 1.** (Continued)

| | STE-OHCA | NSTE-OHCA | p value | Shockable rhythm | Non-shockable rhythm | p value |
|---|---|---|---|---|---|---|
| **S-100b- day 3, μg/l** | 0.111 [0.76–0.208] | 0.129 [0.076–0.243] | 0.690 | 0.109 [0.069–0.181] | 0.183 [0.101–0.349] | 0.571 |

CAD–coronary artery disease; CABG–coronary artery bypass graft; CKD–chronic kidney disease; COPD–chronic obstructive pulmonary disease; CPR–cardiopulmonary resuscitation; eCPR–ECMO-CPR; ECMO–extracorporeal membrane oxygenation; ICD–implantable cardioverter-defibrillator; PAD–peripheral artery disease; PCI–percutaneous coronary intervention; ROSC–return of spontaneous circulation; RRT–renal replacement therapy; TIA–transient ischemic attack.

survival between NSTE-OHCA patients with primarily shockable rhythm receiving PCI (56%) and those not requiring PCI (64%), however, was minor (p = 0.35).

## Discussion

In HACORE, applying a standardized interdisciplinary approach to OHCA patients including routine computed tomography, therapeutic hypothermia and coronary angiography in patients with suspected cardiac cause of arrest [7], we found a high rate of critical coronary stenosis in NSTE-OHCA patients, in particular if NSTE-OHCA patients had a shockable presenting rhythm. More specifically, survival in NSTE-OHCA patients with relevant coronary stenosis receiving early PCI was similar to that observed in STE-OHCA patients and good neurological outcome was observed predominantly in patients with a shockable presenting irrespective of presence or absence of ST-elevations.

**Table 2. Intrahospital parameters and findings of coronary angiography according to ECG findings and initial rhythm.**

| | STE-OHCA | | NSTE-OHCA | | p value | Shockable rhythm | | Non-shockable rhythm | | p value |
|---|---|---|---|---|---|---|---|---|---|---|
| **Coronary angiography (%)** | 180 | (100) | 293 | (87) | <0.001 | 312 | (98) | 161 | (81) | <0.001 |
| **CAD (%)** | | | | | <0.001 | | | | | <0.001 |
| 1-vessel (%) | 60 | (33) | 43 | (15) | | 77 | (25) | 26 | (16) | |
| 2-vessel (%) | 59 | (33) | 37 | (13) | | 72 | (23) | 24 | (15) | |
| 3-vessel (%) | 43 | (24) | 79 | (27) | | 83 | (27) | 39 | (24) | |
| CABG (%) | 3 | (2) | 16 | (5) | | 15 | (5) | 4 | (2) | |
| no sign. CAD (%) | 15 | (8) | 118 | (40) | | 65 | (21) | 68 | (42) | |
| **no PCI (%)** | 29 | (16) | 175 | (60) | | 108 | (35) | 96 | (60) | |
| **PCI (%)** | 151 | (84) | 118 | (40) | <0.001 | 204 | (65) | 65 | (40) | <0.001 |
| **Number of vessels (%)** | | | | | <0.001 | | | | | <0.001 |
| Single (%) | 116 | (77) | 89 | (75) | | 159 | (78) | 46 | (71) | |
| Multiple (%) | 30 | (20) | 25 | (21) | | 38 | (19) | 17 | (26) | |
| missed PCI (%) | 5 | (3) | 4 | (3) | | 7 | (3) | 2 | (3) | |
| **Culprit lesion (%)** | | | | | <0.001 | | | | | <0.001 |
| LAD (%) | 83 | (55) | 51 | (43) | | 104 | (51) | 30 | (46) | |
| LCX (%) | 29 | (19) | 33 | (28) | | 47 | (23) | 15 | (23) | |
| RCA (%) | 36 | (24) | 27 | (23) | | 48 | (24) | 15 | (23) | |
| LMCA (%) | 3 | (2) | 6 | (5) | | 4 | (2) | 5 | (8) | |
| CABG (%) | 0 | (0) | 1 | (1) | | 1 | (0) | 0 | (0) | |

CAD–coronary artery disease; CABG–coronary artery bypass graft; LAD–left anterior descending coronary artery; LCX–left circumflex coronary artery; LMCA–left main coronary artery; PCI–percutaneous coronary intervention; RCA–right coronary artery.

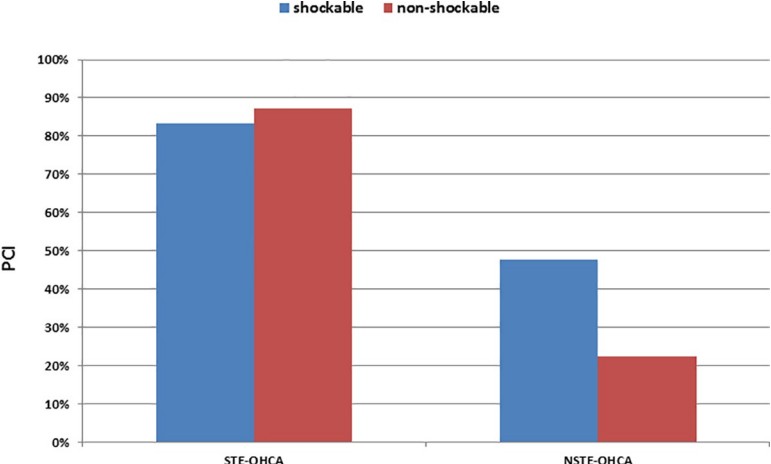

**Fig 1. PCI rates in the HAnnover COoling REgistry (HACORE).** Rate of flow-limiting coronary stenosis in patients following out-of-hospital cardiac arrest (OHCA) is shown depending on the presence of ST-segment-elevations (STE-OHCA) or their absence (NSTE-OHCA) and with respect to the first documented ECG rhythm, shockable (blue) or non-shockable (red).

Our surprisingly high rate of 16% of STE-OHCA patients not requiring PCI might be related to the effort in the emergency ambulance service of acquiring the ECG rapidly. A recent study conducted at three different cardiac arrest centres in Europe reported a rate of false-positive ECGs regarding ST-segment elevations within the first 7 minutes of almost 20% [14]. As we tried to get ECGs written as soon as possible after ROSC, this might have affected our sensitivity in the STE-OHCA group to predict the necessity of PCI.

When reviewing the literature regarding appropriateness of coronary angiography in NSTE-OHCA, in general approximately 30% of patients were reported to have critical coronary stenosis requiring revascularisation [15–17]. In some registries, even OHCA patients with ECGs free of any sign suggesting myocardial ischaemia had impaired coronary flow in 19–33% [17,18]. In an analysis from the Minnesota Resuscitation Consortium, urgent coronary angiography in OHCA patients with primarily shockable rhythm irrespective of ST-elevations was associated with improved outcome. Adjusting for covariates, direct access to the cath

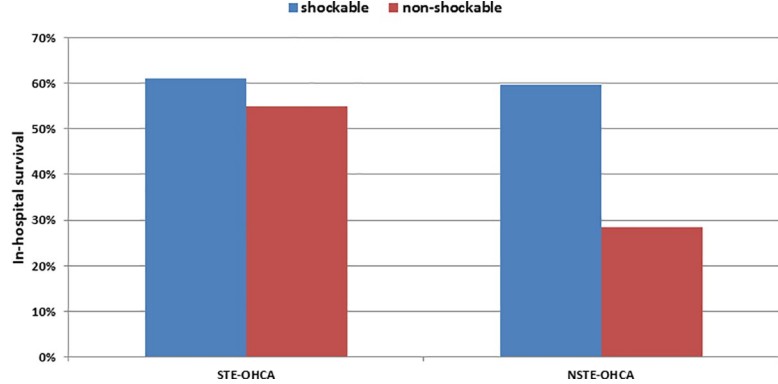

**Fig 2. In-hospital survival in the HAnnover COoling REgistry (HACORE).** In-hospital survival in patients following out-of-hospital cardiac arrest (OHCA) is shown depending on the presence of ST-segment-elevations (STE-OHCA) or their absence (NSTE-OHCA) and with respect to the first documented ECG rhythm, shockable (blue) or non-shockable (red).

lab improved survival with good neurological outcome with an odds ratio of 1.99 (1.07–3.72, p = 0.03), and more specifically NSTE-OHCA patients had an absolute 13% higher survival when treated by protocol with direct cath lab access (adjusted odds ratio 2.77; 1.31–5.85, p = 0.01) [19]. Similarly, we identified the presence of a shockable rhythm as a good indicator for PCI irrespective of ST-segment elevation. PCI in NSTE-OHCA patients was associated with improved outcome [16,19], whereas survival rates in STE-OHCA compared to NSTE-OHCA appeared to be higher in general [17]. Accordingly, we observed a similar survival rate in NSTE-OHCA patients with primary shockable rhythm, which was highly suggestive for a coronary cause of cardiac arrest, as we observed in the group with STE-OHCA. The comparable mortality in NSTE-OHCA patients receiving PCI might be attributable to the rapid access to the cath lab in a standardised patient management as there are no significant delays compared to the access times in STE-OHCA patients. Similarly, in the Minnesota project, early access to coronary angiography with achievement of revascularisation was associated with a three-fold higher probability of survival with a favourable neurological outcome [19]. In a previous retrospective analysis in two large US hospitals, early coronary angiography without PCI was associated with better outcome than not performing coronary angiography in OHCA patients [20]. A meta-analysis of available low-volume data on early coronary angiography in NSTE-OHCA patients supported the use of early coronary angiography in those patients [21]. However, more recently in the prospective, randomised, controlled COACT trial, immediate coronary angiography in NSTE-OHCA patients did not provide a benefit regarding survival of NSTE-OHCA patients [6]. By definition, that trial included only NSTE-OHCA patients if they had shockable rhythms at first ECG and had been haemodynamically stable following ROSC. Furthermore, 65% of the control group received a coronary angiogram on average 5 days after arrest [6]. Of note, only less than one out of three patients in our every-day practice fitted the trials inclusion/ exclusion criteria.

The major reason for not fitting the COACT trials criteria in HACORE was cardiogenic shock following ROSC. We observed cardiogenic shock in 246 out of 337 NSTE-OHCA patients (73%), but PCI was only required in 39% of NSTE-OHCA patients with shock indicating a substantial amount of patients with non-coronary causes of arrest. Nevertheless, survival within the NSTE-OHCA shock group was 42%. Considering that these patients did not have ST-elevations, had all been resuscitated, and were all in shock, a 42% survival rate might represent a reasonable outcome. Many contemporary shock trials in patients with acute myocardial infarction report a mortality rate of 40–50%, whereby the rate of OHCA in those trials was only about 50–60% [22,23]. Even in the very early years of coronary angioplasty there were two predictors of survival: first, successful coronary angioplasty (OR 5.2, p = 0.04), and second absence of need for inotropic drugs (OR 3.6, p = 0.03) [24]. These early findings already strengthen another very important component of early coronary angiography in modern times, the option of access to mechanical circulatory support devices. When analysing all cardiogenic shock patients at our institution we recognized that about two out of three shock patients had OHCA prior to hospital admission [9]. Their chance for survival seemed to be influenced by the availability of early haemodynamic support.

While the COACT trial suggested that early coronary angiography does not provide benefit in NSTE-OHCA patients [6], we realised that our NSTE-OHCA patients included a large group of patients in cardiogenic shock, which is not reflected in that trial. In HACORE, the smaller group of COACT-like patients had a similar in-hospital mortality rate of 31% compared to 35% in the trial.

When putting our data on ECG patterns and angiography results into perspective with previously published observations [15–19,24] and assess the neurological outcome in our patients, it seems not to be the absence or presence of ST-elevations that is predictive for good outcome,

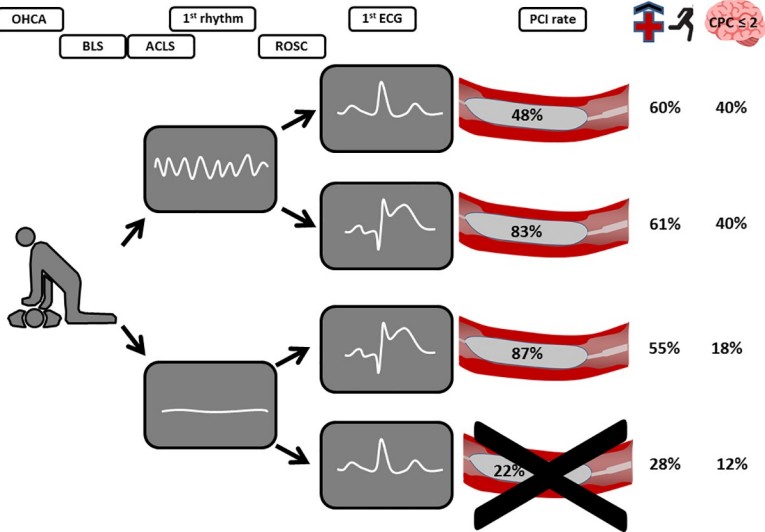

**Fig 3. Rate of coronary stenosis requiring revascularisation, in-hospital survival and good neurological outcome (Cerebral Performance Category (CPC) ≤2) in the HAnnover COoling REgistry (HACORE).**

but rather the primary rhythm that is (Fig 3). Therefore, a standardised approach for all STE-OHCA patients and NSTE-OHCA patients with an initial shockable rhythm appears to be useful, if there is no clear evidence for a non-coronary cause of arrest.

The results observed in our real-life cohort in HACORE are supported by recommendations given by the European Association for Percutaneous Cardiovascular Interventions/ Stent for Life groups regarding invasive coronary angiography in OHCA patients, which recommend immediate coronary angiography in all STE-OHCA and in those NSTE-OHCA patients, who have no obvious non-coronary cause of arrest, no significant comorbidities, and an favourable arrest setting [25]. However, guidelines also recommended considering a stop prior to the cath lab in comatose OHCA survivors without ECG for ST-segment elevation on the post-resuscitation ECG to exclude non-coronary causes of arrest [26].

Similar to the analysis reported here, our previous analysis investigating all patients receiving extracorporeal CPR for ongoing arrest had demonstrated a non-shockable rhythm as a strong factor associated with non-survival [11]. The presence or absence of a shockable rhythm might indicate a potential different etiology of cardiac arrest. Shockable might be more likely to indicate an ischaemic cause, whereas non-shockable might be attributable to anoxic causes. Indeed, baseline characteristics do support such a hypothesis depicting a significantly higher number of chronic pulmonary diseases in patients with non-shockable rhythm and a higher proportion of ST-elevation in shockable rhythm. Patients with non-shockable rhythm had worse conditions of resuscitation indicated by less witnessed arrest and bystander CPR potentially contributing to higher admission lactate and NSE levels (as described in Table 1).

## Limitations

Our registry was performed in a tertiary university hospital setting with a specific algorithm for treating and handling OHCA as well as shock patients. This might influence the results in the way that still comatose patients were aggressively stabilized and treatment was optimized to the local conditions. The data, however, should not be extrapolated to alert patients admitted after short cardiac arrest, who could have a different pattern of underlying disease. While

the sample size of an observational single-centre study has to be considered as a limitation, nevertheless, more than 500 consecutively treated patients are reported.

## Conclusion

Based on the unfavourable outcome and low PCI rate observed in NSTE-OHCA patients with a primary non-shockable ECG rhythm it might be reasonable to restrict routine early coronary angiography to patients with primary shockable rhythms and/or ST-segment elevations after ROSC.

## Supporting information

**S1 Table. Characteristics regarding PCI of NSTE-OHCA and non-shockable patients.** (DOCX)

## Acknowledgments

### Declarations

The authors thank the nursing staff of the catheterization laboratory and cardiology ICU for their continuous support and care in treating OHCA patients.

## Author Contributions

**Conceptualization:** Vera Garcheva, Muharrem Akin, Johann Bauersachs, Andreas Schäfer.

**Data curation:** Vera Garcheva, Muharrem Akin, John Adel, Carolina Sanchez Martinez.

**Formal analysis:** Vera Garcheva, Muharrem Akin, John Adel, Andreas Schäfer.

**Funding acquisition:** Johann Bauersachs.

**Investigation:** Vera Garcheva, Muharrem Akin, John Adel, Andreas Schäfer.

**Methodology:** Muharrem Akin, Carolina Sanchez Martinez, Johann Bauersachs, Andreas Schäfer.

**Project administration:** Johann Bauersachs, Andreas Schäfer.

**Resources:** Andreas Schäfer.

**Supervision:** Muharrem Akin, Johann Bauersachs, Andreas Schäfer.

**Validation:** Vera Garcheva.

**Writing – original draft:** Vera Garcheva, Muharrem Akin, Andreas Schäfer.

**Writing – review & editing:** John Adel, Carolina Sanchez Martinez, Johann Bauersachs.

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
