## [Decision Letter · Decision Letter 0]

8 Feb 2021

PONE-D-21-01798

High Rate of Critical Coronary Stenosis in Patients with Non-ST-Elevation Out-of-Hospital Cardiac Arrest (NSTE-OHCA) – Experience from the HAnnover COoling REgistry (HACORE)

PLOS ONE

Dear Dr. Andreas Schäfer

Thank you for submitting your manuscript to PLOS ONE. After careful consideration, we feel that it has merit but does not fully meet PLOS ONE’s publication criteria as it currently stands. Therefore, we invite you to submit a revised version of the manuscript that addresses the points raised during the review process.

We look forward to receiving your revised manuscript.

Kind regards,

Simone Savastano

Academic Editor

PLOS ONE

Journal Requirements:

2.Please provide additional details regarding participant consent. In the ethics statement in the Methods and online submission information, please ensure that you have specified (1) whether consent was informed and (2) what type you obtained (for instance, written or verbal, and if verbal, how it was documented and witnessed). If your study included minors, state whether you obtained consent from parents or guardians. If the need for consent was waived by the ethics committee, please include this information.

3.We note that you have indicated that data from this study are available upon request. PLOS only allows data to be available upon request if there are legal or ethical restrictions on sharing data publicly. For more information on unacceptable data access restrictions, please see http://journals.plos.org/plosone/s/data-availability#loc-unacceptable-data-access-restrictions.

4.We note that the grant information you provided in the ‘Funding Information’ and ‘Financial Disclosure’ sections do not match.

5.Thank you for stating the following in the Competing Interests section:

"AS received modest lecture fees from ZOLL Inc. regarding therapeutic hypothermia. All other authors have no conflict of interest to declare."

7. Your ethics statement should only appear in the Methods section of your manuscript. If your ethics statement is written in any section besides the Methods, please delete it from any other section.

8.We noticed you have some minor occurrence of overlapping text with the following previous publication(s), which needs to be addressed:

- https://www.jacc.org/doi/full/10.1016/j.jcin.2018.06.022

The text that needs to be addressed involves sentences 1, 2, 8, 9, and 10 of the Introduction. Overlap also should be addressed in Paragraph 2, sentences 3 and 5 of the Discussion.

In your revision ensure you cite all your sources (including your own works), and quote or rephrase any duplicated text outside the methods section. Further consideration is dependent on these concerns being addressed.

Additional Editor Comments (if provided):

Thank you for having submitted your work for evaluation. Your paper covers an important topic with very recent updates which are close to your topic and that you should discuss about. As you will see from the Reviewers' comments the quality and the clarity of the paper need to be improved by following their suggestions.

Reviewers' comments:

Reviewer's Responses to Questions

**Comments to the Author**

1. Is the manuscript technically sound, and do the data support the conclusions?

Reviewer #1: No

Reviewer #2: Partly

2. Has the statistical analysis been performed appropriately and rigorously? 

Reviewer #1: Yes

Reviewer #2: Yes

3. Have the authors made all data underlying the findings in their manuscript fully available?

Reviewer #1: No

Reviewer #2: No

4. Is the manuscript presented in an intelligible fashion and written in standard English?

Reviewer #1: Yes

Reviewer #2: Yes

5. Review Comments to the Author

Reviewer #1: Dear authors,

I believe that your paper is of potential intererst, however I have some concerns that should be addressed and I hope can improve the quality of your manuscript.

Introduction

- your study is focused on post-ROSC ECG and PCI, but you have focused your introduction more on brain and hypotermia. I suggest to re-balance your introduction.

- moreover, especially in the first part of the introduction, many sentence require a reference (e.g. line 58, line 62 and line 64-65).

Methods

- please, specify when you chose, in your hospital, to use ECMO rather than Impella

Results:

- line 163: please, change ECG in rhythm evaluation. ECG here is misleading

- line 168 and line 191: I believe that "when trying to identify potential discriminating factors" should be changed in a more simply "Comparing PCI and non-PCI group..."

- line 176 and line 194: please, specify which are the inclusion criteria of COACT-like and HYPERON-like patients to improve clarity for the readers. Moreover, you should pre-specify these sub-group analysis in the methods

Discussion:

- in general I believe that the discussion should be deeply revised for these reasons:

-- you study is interesting, however it is a observational study and not a RCT, so you have all the typical limitation of the observational studies, including selection bias and a limited sample size. Therefore I believe you cannot state (directly or indirectly) that your study can provide more useful results than a RCT as COACT. The same when comparing Minnesota study with COACT, Minnesota is observational, COACT RCT. I suggest to focus on those sub-groups of patients excluded from the COACT, for example the patients with shock.

-- please, considering all above, mitigate your conclusions

-- I believe that in the discussion you should focus also on the difference regarding survival between shockable and non-shockable (due to different etiology?)

-- you skipped in the discussion an issue that I believe it is very important. About 20% of the STE-patients were not treated with a PCI, this is consistent with previous data in literature and should be stressed. Moreover, considering recent data regarding the key role of timing from ROSC to first ECG acquisition to decrease the percentage of false-positive ECG (JAMA Netw Open. 2021;4(1):e2032875. doi:10.1001/jamanetworkopen.2020.32875), please consider to comment your results in light of this recent evidence

- line 254: "had had" is a typo

Limitations

- please, consider your limited sample size as a limitation. Moreover, your study is observational, therefore this should be recognized in the limitations

- your registry consider only patients in whom hypotermia is performed. This is an important limitation and selection bias and should be highlighted

Reviewer #2: The conclusions expressed in the abstract and in the full paper are slightly different. In the full paper the conclusion is that NSTEMI non-shockable rhythm do not deserve angiography (since just 22% had PCI), while in the abstract is stated that patients with NSTEMI shockable rhythm deserve immediate angiography (since the authors compare the STEMI shockable survival with NSTEMI shockable survival). So if the real conclusions are the ones expressed in the abstract, I think that some information are missing, in particular there is no data about survival in the subgroups NSTEMI shockable PCI vs NSTEMI shockable non PCI. The 60% of survival in NSTEMI shockable patients, which is similar to the STEMI shockable ones (61%), refers to all the NSTEMI shockable (no distinction between PCI vs no PCI), which may be crucial in recommending angiography in this population. Moreover, it's not clear the purpose in citing the HYPERION trial (lines 292-293) in which there is no mention about the link between PCI-rate and in-hospital mortality rate.

6. PLOS authors have the option to publish the peer review history of their article (what does this mean?). If published, this will include your full peer review and any attached files.

Reviewer #1: No

Reviewer #2: No

---

## [Author Response · Author response to Decision Letter 0]

9 Apr 2021

Response to Reviewer #1 Manuscript PONE-D-21-01798

We would like to thank the reviewer for his/her constructive criticism and valuable input. We hope that the changes made to the manuscript and the responses detailed below, satisfactorily address the concerns raised.

Introduction

- your study is focused on post-ROSC ECG and PCI, but you have focused your introduction more on brain and hypothermia. I suggest to re-balance your introduction.

As suggested by the reviewer, we have rewritten the introduction now focusing primarily on post-ROSC-ECG and indication for PCI (lines 61-66).

- moreover, especially in the first part of the introduction, many sentence require a reference (e.g. line 58, line 62 and line 64-65).

The reference to recent AHA ACLS guidelines has been added early on. Most of the other requirements for references are not required anymore as the sentences under concern have been deleted as requested above.

Methods

- please, specify when you chose, in your hospital, to use ECMO rather than Impella

Microaxial pumps were used primarily in isolated left-ventricular failure whereas ECMO was used in refractory arrest or biventricular failure. This has now been added to the patient population section (lines 112-116).

Results:

- line 163: please, change ECG in rhythm evaluation. ECG here is misleading

The wording has been changed as suggested (line 166).

- line 168 and line 191: I believe that "when trying to identify potential discriminating factors" should be changed in a more simply "Comparing PCI and non-PCI group..."

The wording has been changed as suggested (lines 171).

- line 176 and line 194: please, specify which are the inclusion criteria of COACT-like and HYPERON-like patients to improve clarity for the readers. Moreover, you should pre-specify these sub-group analysis in the methods

In the revised version of the manuscript we focused on the COACT trial as it is a trial selectively addressing PCI in NSTE-OHCA patients and we have skipped the reference to the HYPERION trial as also suggested by reviewer #2. Inclusion and major exclusion criteria have been added at the suggested part of the text including the respective reference (lines 180-181). The respective sub-group analysis has now been pre-specified in the Methods section (lines 140-144).

Discussion:

- in general I believe that the discussion should be deeply revised for these reasons:

-- you study is interesting, however it is a observational study and not a RCT, so you have all the typical limitation of the observational studies, including selection bias and a limited sample size. Therefore I believe you cannot state (directly or indirectly) that your study can provide more useful results than a RCT as COACT. 

We have changed the discussion section with regard to this matter. Our intention was not to imply that an observational study provides better results than a RCT. However, observational studies can provide useful information if they indicate that a RCT had been very selective. To this regard, the COACT criteria applied to less than one third of our observational cohort. In other words, every day’s clinical life is much more complex and patients differ largely from the cohort selectively investigated in the trial. Therefore, we believe it is warranted to raise some caution regarding extrapolation of study results into a much more heterogeneous and eventually sicker patient population in every day clinical routine. Following publication of COACT, there was an urge by colleagues that it might generally not be necessary to perform coronary angiography in resuscitated patients if they do not show ST-elevations. However, our registry shows that most patients in clinical routine were not represented by the trial population. Therefore, we tried to find some further indicator to guide patient flow regarding potential beneficiaries from coronary angiography. Following our experience, there is a higher rate of PCI in NSTE-OHCA patients with shockable rhythm; on the contrary, PCI rate and survival in non-shockable rhythms are very low. Therefore, combination of primary rhythm and absence of ST-segment elevations might be a useful indicator to guide patient flow.

The same when comparing Minnesota study with COACT, Minnesota is observational, COACT RCT. I suggest to focus on those sub-groups of patients excluded from the COACT, for example the patients with shock.

-- please, considering all above, mitigate your conclusions

We thank the reviewer for pointing out this important point. However, with all due respect to the reviewer, given the data in our registry it rather appears that the patients in COACT represent a subgroup of the patients being admitted in everyday clinical routine. We never intended to question the trial results; we rather observed a similar outcome in COACT-like patients in HACORE (lines 293-294). Nevertheless, the majority of our patients would have been excluded from the trial mainly due to presence of haemodynamic instability. Considering that these patients did not have STE, all had been resuscitated, and all were in shock, 42% survival is not too bad a result (lines 271-280). 

-- I believe that in the discussion you should focus also on the difference regarding survival between shockable and non-shockable (due to different etiology?)

We agree with the reviewer that the presence or absence of a shockable rhythm might indicate a potential different etiology of cardiac arrest. Shockable might be more likely be indicative for an ischaemic cause, whereas non-shockable might be attributed to anoxic causes. Indeed, baseline characteristics do support such a hypothesis depicting a significantly higher number of chronic pulmonary disease in patients with non-shockable rhythm and a higher proportion of ST-elevation in shockable rhythm. Patients with non-shockable rhythm had worse conditions of resuscitation indicated by less witnessed arrest and bystander CPR potentially contributing to higher admission lactate and NSE levels (lines 312-321).

-- you skipped in the discussion an issue that I believe it is very important. About 20% of the STE-patients were not treated with a PCI, this is consistent with previous data in literature and should be stressed. Moreover, considering recent data regarding the key role of timing from ROSC to first ECG acquisition to decrease the percentage of false-positive ECG (JAMA Netw Open. 2021;4(1):e2032875. doi:10.1001/ jamanetworkopen.2020.32875), please consider to comment your results in light of this recent evidence

We agree with the reviewer on this important point, which has been added to the discussion section. Prior to the cited analysis we intended to perform a 12-channel ECG as rapidly as possible after ROSC (lines 229-235). 

- line 254: "had had" is a typo

We apologize for that, the typo has been corrected; a past perfect does not seem to be necessary. 

Limitations

- please, consider your limited sample size as a limitation. Moreover, your study is observational, therefore this should be recognized in the limitations

This has been done as requested (lines 327-329).

- your registry consider only patients in whom hypothermia is performed. This is an important limitation and selection bias and should be highlighted

With all due respect to the reviewer, we do not believe that strong adherence to guideline recommendations is an important limitation. Our protocol and registry ensure that there is a strict approach towards post-arrest patients in our centre. We just want to confirm to the reviewer that there are no comatose OHCA patients in our hospital with presumed cardiac cause of arrest who are treated by normothermia or no TTM at all. This aspect regarding the setting has been highlighted in detail in the introduction section (lines 82-84). 

Response to Reviewer #2 Manuscript PONE-D-21-01798

We would like to thank the reviewer for his/her constructive criticism and valuable input. We hope that the changes made to the manuscript and the responses detailed below, satisfactorily address the concerns raised.

The conclusions expressed in the abstract and in the full paper are slightly different. In the full paper the conclusion is that NSTEMI non-shockable rhythm do not deserve angiography (since just 22% had PCI), while in the abstract is stated that patients with NSTEMI shockable rhythm deserve immediate angiography (since the authors compare the STEMI shockable survival with NSTEMI shockable survival). So if the real conclusions are the ones expressed in the abstract, I think that some information are missing, in particular there is no data about survival in the subgroups NSTEMI shockable PCI vs NSTEMI shockable non PCI. The 60% of survival in NSTEMI shockable patients, which is similar to the STEMI shockable ones (61%), refers to all the NSTEMI shockable (no distinction between PCI vs no PCI), which may be crucial in recommending angiography in this population. 

We thank the reviewer for pointing out the differing wording which we have corrected in the revised version. However, there is one point in the wording which we would like to strengthen: we used the expression of NSTE-OHCA instead of NSTEMI, because it is very difficult and in many cases impossible after resuscitation to clarify by laboratory parameters alone (e.g. troponin) whether there is an myocardial infarction responsible for release of troponin and the “NSTEMI” is cause of arrest or whether the release of myocardial biomarkers is the consequence of arrest and resuscitation efforts. While patients with NSTEMI with subsequent arrest might have the chance to profit from coronary angiography it is very unlikely that patients with cardiac arrest not related to coronary ischaemia will have a benefit. We have elaborated that problem in the revised introduction section.

As requested, we modified the abstract to be in line with the final conclusion. 

Moreover, it's not clear the purpose in citing the HYPERION trial (lines 293-294) in which there is no mention about the link between PCI-rate and in-hospital mortality rate

We agree with the reviewer that the focus of the HYPERION trial is on therapeutic hypothermia rather than on ECG and PCI and we have, therefore, removed the respective sentences.

---

## [Decision Letter · Decision Letter 1]

16 Apr 2021

PONE-D-21-01798R1

High rate of critical coronary stenosis in patients with Non-ST-elevation out-of-hospital cardiac arrest (NSTE-OHCA) – Experience from the HAnnover COoling REgistry (HACORE)

PLOS ONE

Dear Dr. Andreas Schäfer

Thank you for submitting your manuscript to PLOS ONE. After careful consideration, we feel that it has merit but does not fully meet PLOS ONE’s publication criteria as it currently stands. Therefore, we invite you to submit a revised version of the manuscript that addresses the points raised during the review process.

ACADEMIC EDITOR:

Thank you very much for having addressed the majority of the  reviewers' comments.  However some uncovered issues still remain as highlighted by the reviewers. 

The first is concerning your enrolling only patients who underwent to therapeutic hypothermia. I have no doubt that you apply therapeutic hypothermia  according to guidelines. However, by doing so you did not consider those patients resuscitated from an OHCA and not matching with the hypothermia indications after ROSC (e.g. the awake patients). So my advice for you is  either to acknowledge that as a limitation or to state clearly in the title that you are referring only to hypothermic patients. The second concerns is about the comparison in terms of survival between NSTE patients receiving or not a PCI . The p value required by reviewer#2 is of pivotal importance because some your statement are grounded on that.

We look forward to receiving your revised manuscript.

Kind regards,

Simone Savastano

Academic Editor

PLOS ONE

Journal Requirements:

Additional Editor Comments (if provided):

Thank you very much for having addressed the majority of the reviewers' comments. However some uncovered issues still remain as highlighted by the reviewers.

The first is concerning your enrolling only patients who underwent to therapeutic hypothermia. I have no doubt that you apply therapeutic hypothermia according to guidelines. However, by doing so you did not consider those patients resuscitated from an OHCA and not matching with the hypothermia indications (e.g. not comatose patients). So my advice for you is either to acknowledge that as a limitation or to state clearly in the title that you are referring only to hypothermic patients. The second concerns is about the comparison in terms of survival between NSTE patients receiving or not a PCI . The p value required by reviewer#2 is of pivotal importance because some your statement are grounded on that.

Reviewers' comments:

Reviewer's Responses to Questions

**Comments to the Author**

1. If the authors have adequately addressed your comments raised in a previous round of review and you feel that this manuscript is now acceptable for publication, you may indicate that here to bypass the “Comments to the Author” section, enter your conflict of interest statement in the “Confidential to Editor” section, and submit your "Accept" recommendation.

Reviewer #1: All comments have been addressed

Reviewer #2: (No Response)

2. Is the manuscript technically sound, and do the data support the conclusions?

Reviewer #1: Yes

Reviewer #2: Partly

3. Has the statistical analysis been performed appropriately and rigorously? 

Reviewer #1: Yes

Reviewer #2: Yes

4. Have the authors made all data underlying the findings in their manuscript fully available?

Reviewer #1: No

Reviewer #2: Yes

5. Is the manuscript presented in an intelligible fashion and written in standard English?

Reviewer #1: Yes

Reviewer #2: Yes

6. Review Comments to the Author

Reviewer #1: Dear Authors,

thank you so much for addressing the majority of my concerns.

I have few minor concerns:

- line 55: you stated that "arrhythmias" is a cause of OHCA of cardiac etiology. However, I believe that arrhythmias are the epiphenomen of an underlying heart disease. Or do you mean "primary arrhythmias" such as in LQTS, SdB and so on?

- line 58 and 61: "while urgent"... "while urgent", please rephrase

- line 69-70 "uninterrupted phase of therapeutic hypotermia following ICU admission": there are many demonstrations that coronary angiography can be performed during hypotermia, therefore the decision to perform coronary angiography before ICU admission cannot be based on this justification

Methods:

- line 106: as outlined in my previous revision, I believe that the fact that you included only patients who undergone to hypotermia has to be addressed as a limitation. In fact, following the guidelines, hypotermia is for unconscious patients after ROSC, therefore you excluded all the conscious patients after ROSC, and it is a limitation

Results:

- line 155: you state that 116 OHCA patients required mechanical supporto for cardiogenic shock (Table 1). However, in Table 1 if you add the patients with eCPR, those with Impella and those with vaECMO they are more than 116. Why?

Discussion

- line 226: change "during first rhythm control" in "shockable presenting rhythm"

- line 290-295: I'm aware that the COACT had selective inclusion criteria, but the trial was designed in that way and it is speculative to argue that someone wants to extrapolate the results of a trial (which has, by definition, inclusion and exclusion criteria) to the general population. It is like you argue to extrapolate the results of a trial about the use of Ticagrelor in STEMI patients to all the patients with cardiac ischemic disease.

- line 307-310: I believe that here you introduce a very important point that should be more highlighted, as the importance of a stop in Emergency Department to exclude non-cardiac cause, as reccomanded by 2015 ESC Guidelines on ventricular arrhythmias.

Reviewer #2: Why should routine early PCI be performed on patients with primary shockable rhythms during NSTE-OHCA if it is stated that the difference of survival between NSTE-OHCA patients with primarily shockable rhythm receiving PCI (56%) and those not requiring PCI (64%) was minor (p=??)? It seems that by performing PCI in these patients there is no survival benefit.

7. PLOS authors have the option to publish the peer review history of their article (what does this mean?). If published, this will include your full peer review and any attached files.

Reviewer #1: No

Reviewer #2: No

---

## [Author Response · Author response to Decision Letter 1]

20 Apr 2021

Response to Reviewer #1 Manuscript PONE-D-21-01798R1

We would like to thank the reviewer for his/her constructive criticism and valuable input. We hope that the changes made to the manuscript and the responses detailed below, satisfactorily address the concerns raised.

Dear Authors, thank you so much for addressing the majority of my concerns.

I have few minor concerns:

- line 55: you stated that "arrhythmias" is a cause of OHCA of cardiac etiology. However, I believe that arrhythmias are the epiphenomen of an underlying heart disease. Or do you mean "primary arrhythmias" such as in LQTS, SdB and so on?

Indeed, arrhythmias are in most cases considered an epiphenomenon of underlying heart disease as stated by the reviewer. We have stated this now more clearly as suggested (page 3).

- line 58 and 61: "while urgent"... "while urgent", please rephrase

We have modified the first sentence in order to remove the similar start of subsequent sentences as suggested by the reviewer (page 3).

- line 69-70 "uninterrupted phase of therapeutic hypotermia following ICU admission": there are many demonstrations that coronary angiography can be performed during hypotermia, therefore the decision to perform coronary angiography before ICU admission cannot be based on this justification

We do agree with the reviewer that it is technically feasible to perform coronary angiography during hypothermia. Nevertheless, we wanted to express that we intend to provide consecutive intensive care in general without requiring patients to be transported to other departments for diagnostic procedures such as coronary angiography or computed tomography. As described in Table 1, quite a number of patients are either on renal replacement therapy and/or mechanical circulatory support and unnecessary transports are kept to a minimum. We hope that the revised wording is clearer to understand and not as misleading as the previous one (page 3-4).

Methods:

- line 106: as outlined in my previous revision, I believe that the fact that you included only patients who undergone to hypotermia has to be addressed as a limitation. In fact, following the guidelines, hypotermia is for unconscious patients after ROSC, therefore you excluded all the conscious patients after ROSC, and it is a limitation

We now clarified the fact that all patients were still comatose and receiving hypothermia in the title, the patient population, and the limitations section. (pages 1, 5 & 18).

Results:

- line 155: you state that 116 OHCA patients required mechanical support for cardiogenic shock (Table 1). However, in Table 1 if you add the patients with eCPR, those with Impella and those with vaECMO they are more than 116. Why?

The overall number of patients on mechanical support was 116 as stated in the text. E.g. all eCPR patients are also included in the vaECMO line and several patients received vaECMO plus Impella (n=37 in total), which attributes to the higher number when adding eCPR, Impella and vaECMO. This has now been explained in more detail (page 7).

Discussion

- line 226: change "during first rhythm control" in "shockable presenting rhythm"

This has been changed as suggested (page 13).

- line 290-295: I'm aware that the COACT had selective inclusion criteria, but the trial was designed in that way and it is speculative to argue that someone wants to extrapolate the results of a trial (which has, by definition, inclusion and exclusion criteria) to the general population. It is like you argue to extrapolate the results of a trial about the use of Ticagrelor in STEMI patients to all the patients with cardiac ischemic disease.

We have rephrased that paragraph and now exclude the speculative part just stating that the overall NSTE-OHCA cohort included a large proportion of shock patients (page 16).

- line 307-310: I believe that here you introduce a very important point that should be more highlighted, as the importance of a stop in Emergency Department to exclude non-cardiac cause, as recommended by 2015 ESC Guidelines on ventricular arrhythmias.

We thank the reviewer for this suggestion and have quoted the 2015 ESC guideline as suggested (page 17).

Response to Reviewer #2 Manuscript PONE-D-21-01798

We would like to thank the reviewer for his/her constructive criticism and valuable input. We hope that the changes made to the manuscript and the responses detailed below, satisfactorily address the concerns raised.

Why should routine early PCI be performed on patients with primary shockable rhythms during NSTE-OHCA if it is stated that the difference of survival between NSTE-OHCA patients with primarily shockable rhythm receiving PCI (56%) and those not requiring PCI (64%) was minor (p=??)? It seems that by performing PCI in these patients there is no survival benefit.

We acknowledge this important remark by the reviewer, but due to the observational and non-interventional registry design, we believe that we should not draw this conclusion. We would be allowed to conclude like that, if we had identified NSTE-OHCA patients with primarily shockable rhythms who had relevant coronary stenosis and had them randomized to PCI vs conservative treatment. In our manuscript (and the registry), however, we have the group with relevant coronary stenosis receiving PCI (coronary artery disease as most probable cause of arrest) and the group without coronary stenosis treated conservatively (underlying heart disease with arrhythmic cause of arrest). Therefore, neither can we state that performing PCI provides a benefit nor that “performing PCI in these patients is without survival benefit”. We just compare to very different groups of patients.

Our primary intention was to describe the pattern of underlying disease in a cohort that received routine angiography. To enhance the discussion about the impact on outcome, we have now added the data on good neurological outcome as assessed by cerebral performance category ≤2 in addition to hospital survival and discharge in Figure 3, which now illustrates even more that initial rhythm seems to be more important for outcome than ST-segment elevation. Still, we cannot state that NSTE-OHCA patients receiving PCI have better outcomes than those not receiving PCI. However, we can illustrate that it is not the absence or presence of ST-elevations that is predictive for outcome, but rather the primary rhythm that is. As these patients’ potential outcome is not as futile as many think, we want to encourage the discussion about providing similar care regarding treatment of cardiac ischaemia as it is provided in STE-OHCA patients. This has been added to the discussion on page 16.

Nevertheless, the requested p-value has been added to the sentence on page 13.

---

## [Editor Report · Decision Letter 2]

22 Apr 2021

High rate of critical coronary stenosis in comatose patients with Non-ST-elevation out-of-hospital cardiac arrest (NSTE-OHCA) undergoing therapeutic hypothermia – Experience from the HAnnover COoling REgistry (HACORE)

PONE-D-21-01798R2

Dear Dr. Andreas Schäfer

We’re pleased to inform you that your manuscript has been judged scientifically suitable for publication and will be formally accepted for publication once it meets all outstanding technical requirements.

Kind regards,

Simone Savastano

Academic Editor

PLOS ONE

Additional Editor Comments (optional):

Thank you very much for having addressed also the remaining issues. Now the message you want to deliver with your paper is more clear.
---

## [Editor Report · Acceptance letter]

26 Apr 2021

PONE-D-21-01798R2 

High rate of critical coronary stenosis in comatose patients with Non-ST-elevation out-of-hospital cardiac arrest (NSTE-OHCA) undergoing therapeutic hypothermia – Experience from the HAnnover COoling REgistry (HACORE) 

Dear Dr. Schäfer:

I'm pleased to inform you that your manuscript has been deemed suitable for publication in PLOS ONE. Congratulations! Your manuscript is now with our production department. 

Kind regards, 

on behalf of

Dr. Simone Savastano 

Academic Editor

PLOS ONE